# Resource Screening and Inheritance Analysis of *Fusarium oxysporum* sp. *conglutinans* Race 2 Resistance in Cabbage (*Brassica oleracea* var. *capitata*)

**DOI:** 10.3390/genes13091590

**Published:** 2022-09-04

**Authors:** Long Tong, Cunbao Zhao, Jinhui Liu, Limei Yang, Mu Zhuang, Yangyong Zhang, Yong Wang, Jialei Ji, Bifeng Kuang, Kelan Tang, Zhiyuan Fang, Ryo Fujimoto, Honghao Lv

**Affiliations:** 1Department of Horticulture, Hunan Agricultural University, Changsha 410128, China; 2Key Laboratory of Biology and Genetic Improvement of Horticultural Crops, Institute of Vegetables and Flowers, Chinese Academy of Agricultural Sciences, Ministry of Agriculture and Rural Affairs, Beijing 100081, China; 3Academician and Expert Workstation, Hengyang Vegetable Research Institute, Hengyang 421000, China; 4Graduate School of Agricultural Science, Kobe University, Kobe 657-8501, Japan

**Keywords:** cabbage, Fusarium wilt, resistance identification, race 2, genetic analysis

## Abstract

Cabbage (*Brassica oleracea* var. *capitata*) Fusarium wilt (CFW) is a disease that poses a critical threat to global cabbage production. Screening for resistant resources in order to support the breeding of resistant cultivars is the most reliable approach to control this disease. CFW is caused by *Fusarium oxysporum* f. sp. *conglutinans* (*Foc*), which consists of two physiological races (race 1 and 2). While many studies have focused on resistance screening, gene mining, and inheritance-based research associated with resistance to *Foc* race 1, there have been few studies specifically analyzing resistance to *Foc* race 2, which is a potential threat that can overcome type A resistance. Here, 166 cabbage resources collected from around the world were evaluated for the resistance to both *Foc* races, with 46.99% and 38.55% of these cabbage lines being resistant to *Foc* race 1 and race 2, respectively, whereas 33.74% and 48.80% were susceptible to these two respective races. Of these 166 analyzed cabbage lines, 114 (68.67%) were found to be more susceptible to race 2 than to race 1, and 28 of them were resistant to race 1 while susceptible to race 2, underscoring the highly aggressive nature of *Foc* race 2. To analyze the inheritance of *Foc* race 2 resistance, segregated populations derived from the resistant parental line ‘Badger Inbred 16’ and the susceptible one ‘01-20’ were analyzed with a major gene plus polygene mixed genetic model. The results of this analysis revealed *Foc* race 2-specific resistance to be under the control of two pairs of additive-dominant-epistatic major genes plus multiple additive-dominant-epistatic genes (model E). The heritability of these major genes in the BC_1_P_1_, BC_1_P_2_, and F_2_ generations were 32.14%, 72.80%, and 70.64%, respectively. In summary, these results may aid in future gene mining and breeding of novel CFW-resistant cabbage cultivars.

## 1. Introduction

Cabbage is a cruciferous vegetable that is extensively cultivated throughout the world. Global cabbage yields and associated quality, however, are under persistent threat from cabbage Fusarium wilt (CFW) disease. After first being reported in the State of New York, USA in 1895 [1], CFW has spread rapidly throughout the world affecting major sites of global cabbage cultivation [2,3,4]. After being reported in Yanqing, Beijing in 2001 [5], CFW quickly spread to affect all cabbage cultivation sites in Northern China, resulting in serious losses.

The causative pathogen for CFW is *Foc* [1,6], which consists of two physiological races (Race 1 and 2). While many studies have characterized *Foc* race 1 and associated resistance, and the majority of CFW cases are reportedly caused by race 1, there is evidence that *Foc* race 2 is capable of overcoming type A resistance to *Foc* race 1 [7,8]. Despite such pathogenicity, however, there have been few reports to date specifically focused on *Foc* race 2 or associated CFW resistance.

Chemical and physical approaches are not well suited to controlling the spread of CFW, as *Foc* may remain present in the soil and maintain pathogenicity for over a decade following the initial outbreak. As such, the development and cultivation of CFW-resistance cabbage varieties are generally regarded as the most effective approaches to overcoming this threat, with the mining and screening of resistant lines being central to these resistance breeding efforts. Jones et al. [9] began this resource selection process and identified certain CFW-resistant cabbage varieties including the ‘Wisconsin All Seasons’ and ‘Wisconsin Hollander’ lines. Monteiro and Williams [10] used 23 accessions to test for resistance to several *Brassica* diseases, and the results showed that most of the land races were resistant to CFW. Since these initial discoveries, a series of CFW-resistant varieties have been developed globally. While these cultivars are resistant to CFW caused by *Foc* race 1, few race 2-resistant cultivars have been identified to date.

Genetic analyses of identified resistant resources can provide additional insights to guide further resistance breeding efforts. A major gene plus polygene mixed genetic model is commonly used when analyzing and modeling gene heritability in plants including rice [11], melons [12], tomatoes [13], and cabbages [14]. *Foc* race 1 resistance has previously been identified as a qualitative trait under the control of a single dominant gene (*FOC1*) encoded on chromosome 7 [15,16,17], with the associated resistance being referred to as a type A resistance. In contrast, race 2 resistance is thought to be a quantitative trait under the control of multiple genes under a type B resistance pattern [18,19,20]. To date, however, genetic studies focused on race 2 resistance have been limited. Given the threat posed by *Foc* race 2, there thus remains a persistent need for CFW resistance breeding. Germplasm screening represents an efficient approach to identifying highly resistant and susceptible cabbage lines, providing an opportunity to conduct more detailed analyses of the genetic basis for type B resistance.

The present study was developed to screen for cabbage resources exhibiting resistance to race 2, with 166 accessions collected from throughout the world being analyzed. Subsequent genetic analyses were then performed using the highly CFW-resistant inbred line ‘Badger Inbred 16’ cabbage and the highly susceptible inbred line ‘01-20’ cabbage. The results of these analyses have the potential to provide a set of resources to support future CFW resistance breeding, in addition to aiding in the mapping and cloning of CFW resistance genes.

## 2. Material and Methods

### 2.1. Plant Materials

The inbred line ‘Badger Inbred 16’ (BI-16), which was obtained from the Agricultural Research Service-USA Department of Agriculture (ARS-USDA), exhibits a high degree of resistance to both *Foc* race 1 and race 2 [21,22,23,24]. The line ‘96–100’ was bred through system selection from the hybrid ‘Sheetal’ introduced from India by the Institute of Vegetables and Flowers, Chinese Academy of Agricultural Sciences (IVF-CAAS) in 1996 [25]. The ‘96–100’ exhibits a high degree of resistance to *Foc* race 1 but susceptibilty to race 2. The inbred line ’01-20’ was also bred through system selection from the conventional variety ‘Early Vikings’ introduced from Canada in 1966 by IVF-CAAS. The ‘01-20’ is highly susceptible to both *Foc* race 1 and race 2 [24].

The 166 cabbage lines used in the present study were provided by IVF-CAAS. The inbred lines ’96–100’ and ‘BI-16’ were used as respective resistant controls for *Foc* race 1 and race 2, while the inbred ‘01-20’ line served as a control known to be susceptible to both races. All seedlings were grown in a greenhouse for ~20 days at mean nighttime and daytime temperatures of 20 °C and 28 °C, respectively, until reaching the second-leaf stage. All seedlings were watered 2–3 times per week.

When producing hybrid plants, the line ‘BI-16’ served as the male parent (P_1_) and the one ‘01-20’ as the female parent (P_2_). These two lines were crossed in Spring 2020 to generate F_1_ (P_1_ × P_2_) seeds, with the BC_1_P_1_ [(P_1_ × P_2_) × P_1_]_,_ BC_1_P_2_[(P_1_ × P_2_) × P_2_], and F_2_ [(P_1_ × P_2_)⊗] seeds then being generated via back-crossing and self-crossing performed in 2021, respectively. The resistance of these different populations (P_1,_ P_2,_ F_1,_ BC_1_P_1,_ BC_1_P_2_, and F_2_) to CFW was assessed in the fall, with tested seedlings being cultured as discussed above.

### 2.2. Inoculation and Resistance Testing

The *Foc* race 1 pathogen strain ‘FoAS’ was isolated in 2020 from disease cabbage plants in Anshan, Liaoning province, China, while the race 2 pathogen strain ‘58385’ was obtained from the USA. Inoculation testing was performed via the root-dipping method [26]. All strains were cultured in complete medium (CM) for 3 days at 200 rpm on a rotary shaker (28 °C). Conidial suspensions were adjusted with a hemocytometer to 1 × 10^6^ conidia/mL, after which the roots of seedlings were dipped in this suspension for 15 min. Seedlings were then transferred to plastic pots (10 × 10 × 10 cm) containing sterilized substrate, followed by cultivation at a mean 28 °C temperature in a greenhouse.

The susceptibility of these seedlings to infection was assessed after 10–14 days using previously published scoring standards [24,27,28].

### 2.3. Data Collection and Analysis

Disease index calculations were performed using Microsoft Excel (Microsoft, Redmond, WA, USA), while analyses of the standard deviation and significance values for disease index for the 166 analyzed lines were made using SPSS 20.0 (SPSS, Chicago, IL, USA).

### 2.4. Genetic Analyses

The genetics and heritability of race 2 resistance were analyzed using segregating populations derived from the ‘BI-16’ and ‘01-20’ cabbage lines using a major gene plus polygene mixed genetic model. Maximum likelihood functions and an iterated expectation and conditional maximization (IECM) algorithm were employed when estimating population parameters and frequency distributions. The minimum Akaike information criterion (AIC) value and goodness-of-fit tests including an equal distribution test ($U_{1}^{2}$, $U_{2}^{2}$, and $U_{3}^{2}$), a Smirnov test (_n_W^2^), and a Kolmogorov test (D_n_) were used for optimal model selection [29]. The heritability of major genes and polygenes were then approximated in accordance with the genetic parameters of the optimal selected model.

## 3. Results

### 3.1. Screening for Race 1 and 2 CFW Resistance

In total, 166 cabbage accessions were collected and assessed for their CFW resistance via a root-dipping inoculation approach (Figure 1). The DI values varied markedly from 0–100 among these different cabbage lines for both *Foc* race 1 and 2 (Figure 2, Appendix A). Overall, 34.34% and 27.11% of these accessions were found to be highly resistant to CFW caused by *Foc* race 1 and race 2, respectively, while 12.65% and 11.45% were resistant, 19.28% and 12.65% were moderately resistant, 16.87% and 13.25% were susceptible, and 16.87% and 35.54% were highly susceptible.

To better explore the associations between genotypic characteristics and resistance traits, the maturity, geographic origin, lead color, head shape, and planting season for each of these accessions were analyzed (Table 1). Overall, higher levels of resistance were observed for genotypes introduced from Asia (Japan and Korea) and North America, with 55.14% and 50.00% of these accessions exhibiting *Foc* race 1 resistance, respectively, while 47.67% and 41.67% exhibited *Foc* race 2 resistance. In contrast, only 27.78% and 11.11% of the accessions from China exhibited *Foc* race 1 and race 2 resistance, respectively. Autumnal and overwintering cabbages also exhibited higher levels of resistance relative to spring cabbages for both races, and medium maturity accessions exhibited the highest levels of resistance while mid-late maturity accessions exhibited the greatest susceptibility. Of the 9 medium maturity accessions included in this analysis, 5 and 1 were highly resistant and resistant to race 1, respectively, while 3 and 3 were highly resistant and resistant to race 2. There were no significant differences with respect to resistance rates when comparing flat and round cabbages for either race. As to leaf color, gray-leaved cabbages exhibited the highest levels of resistance, with 5 among 7 accessions being highly resistant and 2 being highly susceptible to race 1, while 4, 1, and 2 were highly resistant, resistant, and highly susceptible to race 2.

Certain differences in DI values were observed when comparing race 1 and race 2 resistance levels. A total of 114 accessions were found to be more susceptible to race 2 relative to race 1, with 28 of these accessions being resistant to race 1 yet susceptible to race 2. Moreover, 41 among 166 accessions were highly resistant to both of these races.

### 3.2. CFW Resistance Frequency Distributions among Segregating Populations

Next, CFW resistance frequency distributions of race 2 were analyzed in segregating populations (Table 2). The average respective DI values for P_1_ and P_2_ were 0 and 92.00, with the value for the F_1_ population (35.45) thus being lower than the mean value for these two parental lines (46.00) and more like that for P_1_. This suggests that CFW resistance is subject to partial dominance for this breeding combination. Average DI values for the BC_1_P_1_, BC_1_P_2_, and F_2_ populations were 20.13, 54.83, and 45.97, respectively. CFW resistance frequency distributions in the BC_1_P_1_, BC_1_P_2_, and F_2_ populations revealed multiple peaks in both the BC_1_P_2_ and F_2_ populations as well as a skewed BC_1_P_2_ population distribution, consistent with the genetic characteristics of quantitative traits (Figure 3).

### 3.3. Optimal Genetic Model Selection and Testing

Next, a major gene plus polygene mixed genetic model for quantitative traits was employed to analyze *Foc* race 2 resistance in these cabbage cultivars. Maximum likelihood function and AIC values were thus generated for 23 genetic models (Table 3), with these models then being grouped into 5 categories: A (one major gene); B (two major genes); C (polygene); D (one major gene plus a polygene); and E (two major genes plus a polygene).

Minimum AIC values were next used to select the three most promising candidate models, which included models E, E-1, and E-3. These models were then subjected to goodness-of-fit testing (Table 4), revealing that 11, 12, and 13 values for models E, E-1, and E-3, respectively, reached significance levels. As such, model E was identified as the most optimal model, indicating that CFW resistance was under the control of two pairs of additive-dominant-epistatic major genes plus multiple additive-dominant-epistatic genes.

### 3.4. Genetic Parameter Estimations

Through a least-squares approach, first-order and second-order parameters for model E were estimated next (Table 5). First-order parameter analyses indicated that the respective additive effects of the major genes (h_a_ and h_b_) were −1.25 and −1.15, indicating that they contribute to the weakening of resistance. The dominant effect and potential ratios for the first major gene were −0.78 and 0.62, respectively, while for the second major gene they were 0.29 and −0.26. This indicated that the first major gene exhibited partial dominance, with the degree of dominance being significantly higher than that for the second major gene, which exhibited negative partial dominance with a low degree of dominance. The respective epistatic effects for additive × additive (i), additive × dominant (j_ab_), dominant × additive (j_ba_) and dominant × dominant (l) interactions were 0.32, −1.07, 0.98, and 2.12, consistent with an interaction between these two major genes.

Second-order parameter analyses revealed that the heritability values for these major genes in the BC_1_P_1_, BC_1_P_2_, and F_2_ populations were 32.14%, 72.80%, and 70.64%, respectively (Table 5), while the respective heritability values for multiple genes in these populations were 49.47, 18.13, and 19.57%. Major genes thus exhibited significantly greater heritability than did minor genes in the BC_1_P_2_ and F_2_ populations, although the opposite was true in the BC_1_P_1_ population, thus indicating that minor genes can impact CFW resistance. Environmental factors also exhibited an effect, with the BC_1_P_1_, BC_1_P_2_, and F_2_ populations exhibiting respective variation values of 18.38, 9.07, and 9.79.

## 4. Discussion

Since first being detected in the USA, CFW has emerged as a leading threat to global cabbage production, resulting in major crop losses and economic damage. Owing to its soil-borne nature, physical or chemical approaches to preventing the spread of this disease remain largely ineffective [30,31]. Breeding and cultivation of CFW-resistant cabbage varieties is thought to represent the most economical and effective approach to overcoming CFW. New resistant varieties can be identified through resistance screening efforts, thus forming the basis for subsequent disease-resistant breeding.

At present, Fusarium wilt resistance is primarily identified in plants during the seedling stage, with root-dipping being the most commonly employed strategy given that it most closely mimics the route whereby plants are exposed to this pathogen in nature, with this approach having been implemented in cotton [32], watermelons [33], bananas [34], and beans [35]. This same approach was thus employed in the present study. Given the global prevalence and severity of CFW, several cabbage cultivars exhibiting type A resistance to race 1 have been generated in recent decades, although race 2 can still affect many of these cabbage varieties [8,36]. Jones et al. [9] were the first to obtain disease-resistant varieties from resource screening.

Consistently, 114/166 cabbage lines analyzed in the present study were found to be more susceptible to race 2 relative to race 1; 28 lines were resistant to race 1 while susceptible to race 2, whereas 41 lines were highly resistant to both of these races, respectively, accounting for 71.9% and 89.1% of the resistant materials. Given the high degree of coincidence between resistance to these two *Foc* races, this may suggest that there is some genetic relationship between type A and type B resistance. One possibility may be that the race 1 resistance gene *FOC1* [15] can also contribute to race 2 resistance, although further research will be needed to assess this possibility directly. In addition, cabbage lines originating from Japan, Korea, and America were more resistant than those lines derived from other locations, potentially because CFW was studied at an earlier time point in these nations, leading to the more extensive screening and breeding of CFW-resistant cabbage cultivars. As such, the introduction of more cabbage varieties from these countries is warranted to support global resistance breeding efforts. Moreover, those germplasms with a gray leaf color exhibited a higher resistance ratio, potentially owing to the higher levels of wax and epidermal cuticle thickness exhibited by these varieties, suggesting that these properties may protect against CFW or reduce its severity.

Genetic analyses are central to the effective breeding of disease-resistant plants. As discussed above, Walker et al. [20,37] initially defined two forms of CFW resistance, with type A resistance to *Foc* race 1 being under the control of one dominant gene that has since been cloned successfully [15,17], whereas type B resistance to *Foc* race 2 is polygenic. The results of the present study indicated that race 2 resistance is under the control of two pairs of additive-dominant-epistatic major genes plus additive-dominant-epistatic multiple genes (model E), in line with previous research while successfully expanding on these prior results by providing a more detailed genetic overview of the basis for *Foc* race 2 resistance and associated genetic parameters. Together, these data will provide a theoretical foundation for future efforts to breed CFW-resistant cabbage cultivars.

## 5. Conclusions

In summary, this study leveraged 166 cabbage accessions to explore the prevalence and characteristics of CFW resistance. Overall, 34.34% and 27.11% of these lines were found to be highly resistant to CFW caused by *Foc* race 1 and *Foc* race 2, respectively, while 12.65% and 11.45% were resistant, 19.28% and 12.65% exhibited intermediate resistance, and the remaining lines were either susceptible or highly susceptible to these diseases. The aggressive nature of race 2 was underscored by the fact that 114 cabbage lines exhibited greater susceptibility to race 2 relative to race 1. In addition, 41 lines were highly resistant to both race 1 and race 2. Subsequent analyses of the heritability of race 2 resistance were conducted using segregating populations derived from the ‘BI-16’ and ‘01-20’ parental lines, which were, respectively, highly resistant and highly susceptible to race 2. These results revealed that race 2 resistance was under the control of two pairs of additive-dominant-epistatic major genes plus multiple additive-dominant-epistatic genes (model E). The heritability of the major genes in the BC_1_P_1_, BC_1_P_2_, and F_2_ generations was 32.14%, 72.80%, and 70.64%, respectively. Together, these results highlight a robust resource set that can provide a valuable foundation for future CFW resistance-focused research.

## Figures and Tables

**Figure 1 genes-13-01590-f001:**
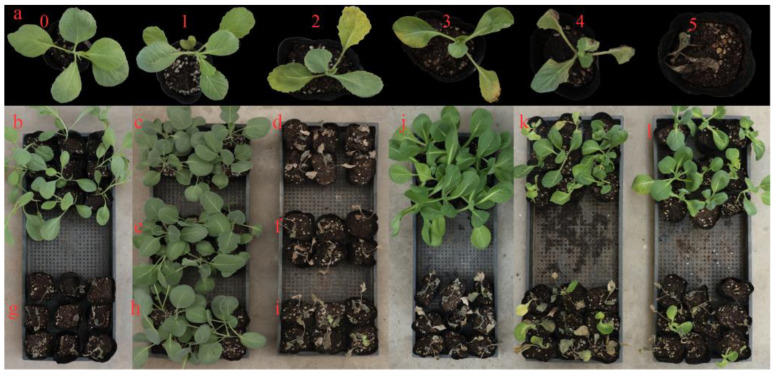
The disease grades of leaves and resistance performance of different materials. (**a**) Disease degrades of levels. 0: no symptoms; 1: slight yellowing of one leaf; 2: moderate yellowing of 1–2 leaves; 3: severe yellowing or wilting of at least half of leaves; 4: severe yellowing or wilting of all leaves other than the core leaves; 5: severe yellowing or wilting of all leaves, or plant death. (**b**-**i**) resistance performance of different materials to race 2. (**b**) ‘BI-16’ (resistant control). (**c**) ‘HB1186’. (**d**) ‘21-3’. (**e**) ‘YF’. (**f**) ‘23202’. (**g**) ‘01-20’ (susceptible control). (**h**) ‘XQ’. (**i**) ‘01-88’. (**j**–**l**) The same materials were resistant to race 1 while susceptible to race 2. (**j**) ‘JTM’. (**k**) ‘CF3’. (**l**) ‘MYF’.

**Figure 2 genes-13-01590-f002:**
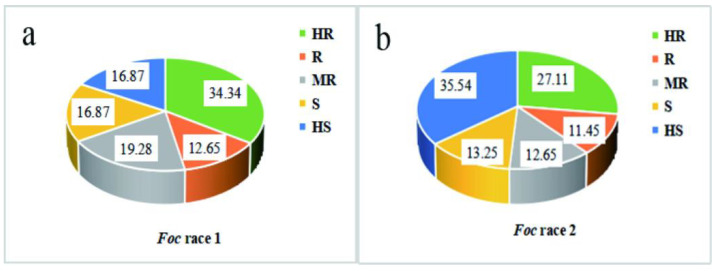
Proportion of resistance levels of different cabbage materials to *Foc*. (**a**,**b**) represent the proportion of materials with different resistance to *Foc* race 1 and 2. HR = highly resistant; R = resistant; MR = moderately resistant; S = susceptible; HS = highly susceptible.

**Figure 3 genes-13-01590-f003:**
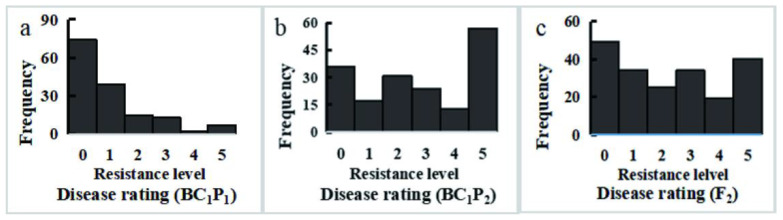
Frequency distribution of the CFW resistance in BC_1_P_1_, BC_1_P_2_, and F_2_. (**a**) BC_1_P_1_, (**b**) BC_1_P_2_, and (**c**) F_2_ populations of ‘BI-16’ × ‘01-20’.

**Table 1 genes-13-01590-t001:** The relationship between genotypes and resistance.

		Race 1						Race 2					
		Number of Accessions	Percentage of Plants (%)					Number of Accessions	Percentage of Plants (%)				
Classification			Highly Resistant	Resistant	Moderately Resistant	Susceptible	Highly Susceptible		Highly Resistant	Resistant	Moderately Resistant	Susceptible	Highly Susceptible
Geographic origin	China	18	5.56	22.22	27.78	16.67	27.78	18	5.56	5.56	11.11	22.22	55.56
	Asia except China	107	42.06	13.08	17.76	14.95	12.15	107	35.51	12.15	11.21	12.15	28.97
	North America	12	41.67	8.33	16.67	8.33	25.00	12	33.33	8.33	8.33	16.67	33.33
	Europe	29	20.69	6.90	20.69	27.59	24.14	29	6.90	13.79	20.69	10.34	48.28
Planting season	Spring	43	23.26	4.65	27.91	20.93	23.26	43	11.63	6.98	13.95	20.93	46.51
	Autumn	90	40.00	14.44	15.56	14.44	15.56	90	32.22	11.11	12.22	12.22	32.22
	Overwintering	33	33.33	18.18	18.18	18.18	12.12	33	33.33	18.18	12.12	6.06	30.30
Maturity	Early maturity	53	39.62	7.55	20.75	13.21	18.87	53	24.53	11.32	11.32	11.32	41.51
	Mid-early maturity	38	34.21	10.53	18.42	18.42	18.42	38	28.95	10.53	13.16	18.42	28.95
	Medium maturity	9	55.56	11.11	22.22	11.11	0.00	9	33.33	33.33	11.11	11.11	11.11
	Mid-late maturity	12	25.00	0.00	25.00	25.00	25.00	12	25.00	0.00	16.67	16.67	41.67
	Late maturity	54	27.78	22.22	16.67	20.37	12.96	54	27.78	11.11	12.96	11.11	37.04
Head shape	Flat	59	28.81	18.64	22.03	16.95	13.56	59	28.81	11.86	13.56	13.56	32.20
	Round	107	37.38	9.35	17.76	17.76	17.76	107	26.17	11.21	12.15	13.08	37.38
Leaf color	Grey	7	71.43	0.00	0.00	0.00	28.57	7	57.14	14.29	0.00	0.00	28.57
	Gray green	67	33.33	15.79	21.05	17.54	12.28	67	29.82	12.28	12.28	33.33	12.28
	Green	64	28.13	17.19	17.19	17.19	20.31	64	18.75	14.06	15.63	14.06	37.50
	Dark green	38	39.47	2.63	23.68	18.42	15.79	38	31.58	5.26	10.53	10.53	42.11

**Table 2 genes-13-01590-t002:** Frequency distribution of CFW resistance levels in segregated populations derived from ‘BI-16’ and ‘01-20’.

Generation	Number	Frequency Distribution of FW Disease Rating in Each Population						Mean Disease Index
		Level 0	Level 1	Level 2	Level 3	Level 4	Level 5	
P_1_ (BI-16)	15	15	0	0	0	0	0	0.00
P_2_ (01-20)	15	0	0	0	1	6	8	92.00
F_1_ (BI-16 × 01-20)	22	4	6	5	6	0	1	35.45
BC_1_P_1_ (BI-16 × 01-20 × BI-16)	150	74	39	15	13	2	7	20.13
BC_1_P_2_ (BI-16 × 01-20 × 01-20)	178	36	17	31	24	13	57	54.83
F_2_ (BI-16 × 01-20) ⊗	201	49	34	25	34	19	40	45.97

**Table 3 genes-13-01590-t003:** The estimation of the maximum likelihood values and AIC values of the genetic model.

Model	Implication of Model	Maximum Likelihood Value	AIC
A-1	1 MG-AD	−1073.03	2154.06
A-2	1 MG-A	−1110.42	2226.84
A-3	1 MG-EAD	−1082.03	2170.05
A-4	1 MG-AEND	−1156.68	2319.36
B-1	2MG-AD1	−1023.43	2066.86
B-2	2MG-AD	−1070.37	2152.73
B-3	2MG-A	−1165.04	2338.08
B-4	2MG-EA	−1123.04	2252.09
B-5	2MG-AED	−1081.75	2171.50
B-6	2MG-EEAD	−1105.63	2217.25
C	PG-ADI	−1101.32	2222.63
C-1	PG-AD	−1106.42	2226.85
D	MX1-AD-ADI	−1103.08	2230.17
D-1	MX1-AD-AD	−1096.19	2210.38
D-2	MX1-A-AD	−1096.19	2196.55
D-3	MX1-EAD-AD	−1081.53	2179.06
D-4	MX1-AEND-AD	−1101.55	2219.10
E *	MX2-ADI-ADI	−988.78	2013.56
E-1 *	MX2-ADI-AD	−1009.09	2048.19
E-2	MX2-AD-AD	−1055.47	2132.94
E-3 *	MX2-A-AD	−1013.12	2044.25
E-4	MX2-EAED-AD	−1106.06	2228.11
E-5	MX2-AED-AD	−1054.71	2127.42

Note: * represents the candidate model selected based on their smaller AIC values. MG: Major gene model; PG: Polygene model; MX: Mixed major gene and polygene model; A: Additive effect; D: Dominant effect; E: Equal. I: Interaction (epistasis); N: Negative.

**Table 4 genes-13-01590-t004:** Tests for goodness of fit model of CFW resistance in segregated generations.

Model	Generation	$U_{1}^{2}$	$U_{2}^{2}$	$U_{3}^{2}$	_n_W^2^	D_n_
E	P_1_	0.00 (1.00)	1.17 (0.28)	18.75 (0.00) *	1.05 (<0.05) *	0.40 (<0.05) *
	P_2_	0.05 (0.83)	1.40 (0.24)	30.82 (0.00) *	0.45 (>0.05)	0.26 (>0.05)
	F_1_	0.09 (0.76)	0.34 (0.56)	1.35 (0.25)	0.15 (>0.05)	0.26 (>0.05)
	BC_1_P_1_	5.03 (0.02) *	5.12 (0.02) *	0.13 (0.72)	2.02 (<0.05) *	0.21 (<0.05) *
	BC_1_P_2_	3.51 (0.06)	1.84 (0.17)	3.33 (0.07)	1.21 (<0.05) *	0.08 (>0.05)
	F_2_	3.30 (0.07)	4.03 (0.04) *	0.99 (0.32)	0.86 (<0.05) *	0.67 (>0.05)
E-1	P_1_	12.95 (0.00) *	13.19 (0.00) *	0.35 (0.55)	2.13 (<0.05) *	0.67 (<0.05) *
	P_2_	4.55 (0.03) *	0.54 (0.46)	28.27 (0.00) *	0.79 (<0.05) *	0.34 (>0.05)
	F_1_	1.81 (0.18)	1.41 (0.23)	0.21 (0.65)	0.25 (>0.05)	0.26 (>0.05)
	BC_1_P_1_	0.21 (0.65)	0.78 (0.38)	3.14 (0.08)	1.62 (<0.05) *	0.15 (<0.05) *
	BC_1_P_2_	2.54 (0.11)	1.47 (0.23)	1.76 (0.19)	0.99 (<0.05) *	0.07 (<0.05) *
	F_2_	0.15 (0.70)	0.01 (0.92)	1.29 (0.26)	0.47 (>0.05)	0.03 (<0.05) *
E-3	P_1_	17.55 (1.00)	14.99 (0.00) *	0.54 (0.46)	2.51 (<0.05) *	0.71 (<0.05) *
	P_2_	6.04 (0.01) *	1.18 (0.28)	26.75 (0.00) *	0.89 (<0.05) *	0.34 (>0.05)
	F_1_	2.04 (0.15)	1.70 (0.19)	0.11 (0.75)	0.27 (>0.05)	0.26 (>0.05)
	BC_1_P_1_	0.09 (0.76)	0.47 (0.49)	2.50 (0.11)	1.59 (<0.05) *	0.15 (>0.05)
	BC_1_P_2_	2.64 (0.10)	1.38 (0.24)	2.57 (0.11)	0.99 (<0.05) *	0.07 (<0.05) *
	F_2_	6.21 (0.01) *	5.08 (0.02) *	0.41 (0.52)	1.12 (<0.05) *	0.08 (<0.05) *

Note: $U_{1}^{2}$, $U_{2}^{2}$ and $U_{3}^{2}$ represents the statistics of the uniformity test; _n_W^2^ represents the statistic of the Smirnov test; D_n_ represents the statistic of the Kolmogorov test. The probability of $U_{1}^{2}$, $U_{2}^{2}$ and $U_{3}^{2}$ is presented in parentheses; the threshold limit of _n_W^2^ at the 0.05 level is 0.461; * indicates significance at the 0.05 level.

**Table 5 genes-13-01590-t005:** The estimation of genetic parameters of fit a model of CFW resistance.

First-Order Parameter	Estimate	Second-Order Parameter	Estimate		
			B_1_	B_2_	F_2_
m_1_	2.08	σ_mg_^2^	0.58	2.67	2.40
m_2_	3.00	σ_pg_^2^	0.89	0.66	0.66
m_3_	1.74	σ_p_^2^	1.81	3.66	3.39
m_4_	2.54	σ_e_^2^	0.33	0.33	0.33
m_5_	2.02	h_mg_^2^	32.14	72.80	70.64
m_6_	1.96	h_pg_^2^	49.47	18.13	19.57
d_a_	−1.25	1 − (h_mg_^2^ + h_pg_^2^)	18.38	9.07	9.79
d_b_	−1.15				
h_a_	−0.78				
h_b_	0.29				
h_a_/d_a_	0.62				
h_b_/d_b_	−0.26				
i	0.32				
j_ab_	−1.07				
j_ba_	0.98				
l	2.12				

Note: The subscripts a and b refer to two major genes; m: population mean; d_a_: additive effect of the first major gene; d_b_: additive effect of the second major gene; h_a_: dominant effect of the first major gene; h_b_: dominant effect of the second major gene; i: epistatic effect value of additive × additive between d_a_ and d_b_; j_ab_: epistatic effect value of additive × dominant between d_a_ and h_b_; j_ba_: epistatic effect value of dominant × additive between h_a_ and d_b_; l: epistatic effect value of dominant × dominant between h_a_ and h_b_; σ_p_^2^: phenotypic variance; σ_pg_^2^: polygene variance; σ^2^: environmental variance; σ_mg_^2^: major gene variance; h_mg_^2^: major gene heritability; h_pg_^2^: polygene heritability; 1 − (h_mg_^2^ + h_pg_^2^): Environmental variance.

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
