# Peer review of "Resource Screening and Inheritance Analysis of Fusarium oxysporum sp. conglutinans Race 2 Resistance in Cabbage (Brassica oleracea var. capitata)"

_genes, 2022, doi:10.3390/genes13091590_

Round 1

Reviewer 1 Report

Authors performed study aiming at characterization of cabbage inheritance developent o Fusarium oxysporum race 2 infection. Obtained results improve the understanding of genetic mechanism leading to plant resistance development. Study is well planned and realized. Obtained data support conclusions. Minor comments should be addressed.

Line 56; add space after 2001.

Section 2.1 Why Authors selected particular plant lines? Did they indicate particularly high values of resistance or susceptibility?

Section 2.2; Why Authors used presented two pathogen strains?

Line 166; using lines instead of accessions could be more clear.

Line 173; correct spaces inside the bracket.

Sentence in lines 210-214 should be rewritten to be more clear. Instead of 5/7 could be 5 among 7. Similarly line 219. Instead of 41/166 could be 41 among 166.

Line 235; in the text, the DI for P2 is 92.00, while in the table 2 it is 89.33. Correct it.

Line 240; in the text, the DI for F2 is 45.92, while in the table 2 it is 45.97. Correct it.

Line 265; Model E indicates 11 statistically significant values, while E3 shows 13. Why the E model was selected, not the E3?

Sentence in line 328/329 should be corrected.

Authors could add in Discussion, for example based on available references, which genes could be responsible for resistance to Fusarium oxysporum race 2 infection.  

All Supplementary data should be checked and added to the main body of manuscript.

Author Response

Thank you for your review and valuable comments on our article.

  1. We have added space after 2001.
  2. As we said in the section 2.1, the three lines we selected were proved to be resistant or susceptible to Foc. The line ‘Badger Inbred 16’is resistant to both of the two races, line ‘96-100’ is resistant to race 1 while susceptible to race 2, and line ‘01-20’ is susceptible to both of the two races.
  3. There are two pathogens that could cause cabbage cabbage Fusarium wilt, named race 1 and race 2. The two stains we selected represent the two races.
  4. We have revised as you suggested.
  5. We have checked and corrected the data in line 240 and table 2.
  6. In the goodness-of-fittest, more significant values mean the model is less reliable.
  7. As we said in the Discussion, we suspected that FOC1may affect the resistance of race 2.
  8. The Supplementary data has been checked again.

Reviewer 2 Report

The manuscript is very well written and 

Line 49

replace "tested" with "test"

Author Response

Thank you for your review and valuable comments on our article.

  1. We have corrected as you suggested.
